# Amino acid gas phase circular dichroism and implications for the origin of biomolecular asymmetry

Cornelia Meinert [1✉], Adrien D. Garcia[1,5], Jérémie Topin [1,5], Nykola C. Jones [2], Mira Diekmann[3], Robert Berger[3], Laurent Nahon [4], Søren V. Hoffmann [2] & Uwe J. Meierhenrich [1✉]

Life on Earth employs chiral amino acids in stereochemical L-form, but the cause of molecular symmetry breaking remains unknown. Chiroptical properties of amino acids – expressed in circular dichroism (CD) – have been previously investigated in solid and solution phase. However, both environments distort the intrinsic charge distribution associated with CD transitions. Here we report on CD and anisotropy spectra of amino acids recorded in the gas phase, where any asymmetry is solely determined by the genuine electromagnetic transition moments. Using a pressure- and temperature-controlled gas cell coupled to a synchrotron radiation CD spectropolarimeter, we found CD active transitions and anisotropies in the 130–280 nm range, which are rationalized by ab initio calculation. As gas phase glycine was found in a cometary coma, our data may provide insights into gas phase asymmetric photochemical reactions in the life cycle of interstellar gas and dust, at the origin of the enantiomeric selection of life's L-amino acids.

[1] Institut de Chimie de Nice, Université Côte d'Azur, UMR 7272 CNRS, 06108 Nice, France. [2] ISA, Department of Physics and Astronomy, Aarhus University, 8000 Aarhus, Denmark. [3] Fachbereich Chemie, Philipps-Universität Marburg, 35032 Marburg, Germany. [4] Synchrotron SOLEIL, L'Orme des Merisiers, 91192 Gif-sur-Yvette, France. [5] These authors contributed equally: Adrien D. Garcia, Jérémie Topin. ✉email: cornelia.meinert@univ-cotedazur.fr; uwe.meierhenrich@univ-cotedazur.fr

Evolution has led to proteins with an incredible array of structures and functions. All proteins have in common that their molecular structure exclusively uses amino acids in stereochemical L-configuration. However, the origin of the molecular asymmetry and the conformational landscape of the 21 left-handed proteinogenic amino acids that orchestrate the assembly of proteins into their unique, biologically relevant three-dimensional structures[1], is still not properly understood. The importance of studying isolated pure amino acids in the gas phase becomes apparent, not only in terms of changes to their conformation due to (i) the absence of the polarizability of the surrounding medium[2], (ii) intermolecular hydrogen bonding or coordination with adjacent molecules, but also (iii) because amino acids exist as zwitterions in solution and solid state[3], whereas they are in the non-ionic molecular form when isolated in the gas phase[4,5]—the configuration amino acids adopt in proteins and polypeptides[6] as well as in interstellar gas clouds.

In the last few decades, experimental spectroscopic approaches using electronic or vibrational spectroscopy[7–11], microwave and millimeter wave spectroscopy[12–15], as well as photoelectron circular dichroism (PECD)[16,17] for probing the structural properties of amino acids in the gas phase collectively aimed to map the conformational behavior of protein-embedded amino acids. However, absorption-based chiroptical measurements in the gas phase have so far been inaccessible for amino acids because the target molecules show insufficient vapor pressure leading to low gas phase density and therefore to poor signal-to-noise ratios. The very few efforts made, for instance via infrared Cavity-Ring Down Spectroscopy (CRDS)[18] to probe circular birefringence and circular dichroism in the gas phase, have so far been limited to volatile organic species not including amino acids[19]. First attempts in the VUV on alanine, even with the very-sensitive technique of ion yield, provided only upper limits without succeeding in providing the absolute configuration[20]. With a specifically constructed gas cell (Fig. 1) coupled to a synchrotron spectropolarimeter to lower the beam divergence that occurs with increased path length (Supplementary Methods), we have now recorded the CD and anisotropy spectra for enantiomeric pairs of gas phase amino acids including alanine (Fig. 2), 2-aminobutyric acid, proline, valine, norvaline, isovaline, and leucine (Supplementary Note 1) in the vacuum-ultraviolet and far UV regions.

## Results

**Gas phase chiroptical spectra.** The gas cell with an optical path of 500 mm (Fig. 1) allowed constant elevated temperatures (423–473 K) to be maintained, at which sufficient vapor pressures of our target molecules enabled reproducible measurements of the chiroptical properties (Supplementary Note 2), while minimizing thermal decomposition (Supplementary Note 3). We obtained highly consistent data on pure gas phase molecules without chiroptical secondary effects such as those linked to the first polarized solvation shell, chiral induction effects, and local microcrystallization, as observed in condensed phases. Solvent cut-off effects did not limit the energetic range of these spectra. The L-alanine enantiomer shows negative CD states at $\lambda = 134$, 174, and 236 nm; positive CD states at 152 and 205 nm. D-alanine shows the expected mirroring effect (Fig. 2).

The anisotropy spectrum, which is the absorbance $(A)$ normalized CD spectrum, $g(\lambda) = CD(\lambda)/A(\lambda)$, is directly related to the enantiomeric excess $(ee)$ inducible by enantioselective photolysis[21]. Compared to the CD spectrum, the anisotropy spectrum has enhanced intensity at long wavelength, despite the much more intense CD states at shorter wavelengths. Our data show that highest enantiomeric excesses can be photochemically induced into alanine at $\lambda = 214$ nm and $\lambda = 244$ nm.

An overview of gas phase CD and anisotropy spectra of the investigated D-amino acids and L-amino acids is given in Table 1 and Supplementary Table 1, with full spectra shown in the supplementary information. Both the sign and the magnitude of the gas phase CD and anisotropy bands are shown to be sensitive to the lateral side chain of amino acids (Fig. 2c). Highest correlation in the position and magnitude of CD bands are observed for the three straight-chain amino acids alanine, 2-aminobutyric acid, and norvaline. Methyl group substitution at the alpha carbon, as in the non-coded amino acid isovaline, led to sign inversion of almost all transitions. The unique five-membered nitrogen heterocycle moiety in proline resulted in hypsochromic shifts of the two lowest energy transitions, as well as different spectral features in the high-energy region compared to all other amino acids investigated. Chiroptical spectra of serine could not be measured because enantiomers decomposed upon sublimation, while phenylalanine showed very high absorbance but very weak CD bands and consequently weak anisotropies in the lower $10^{-4}$ range.

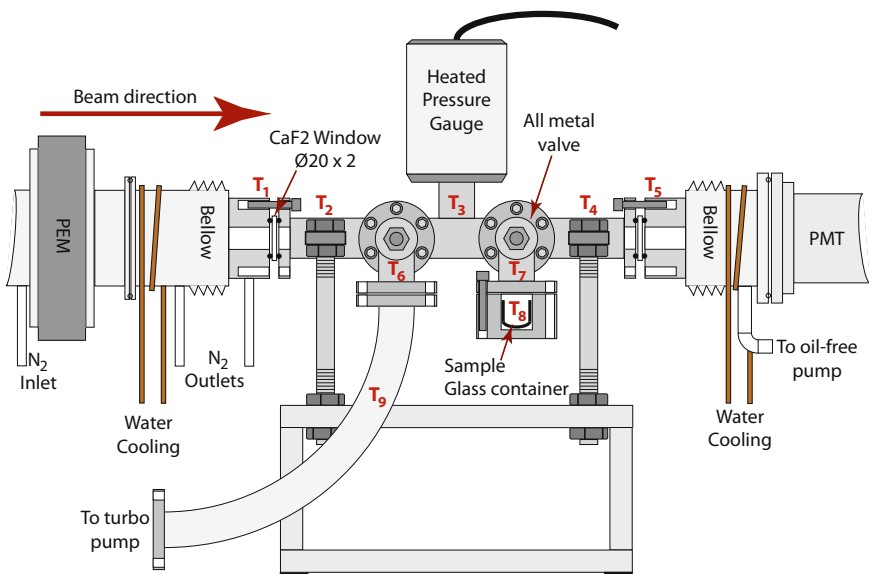

**Fig. 1 Temperature and pressure-controlled gas cell providing sufficient molecular density to perform gas phase CD and anisotropy spectroscopy.** The photo elastic modulator (marked PEM) produces alternating left-handed and right-handed polarized light. The detector is a photomultiplier tube (marked PMT). The nine different temperature monitor and control points are labeled T1–T9.

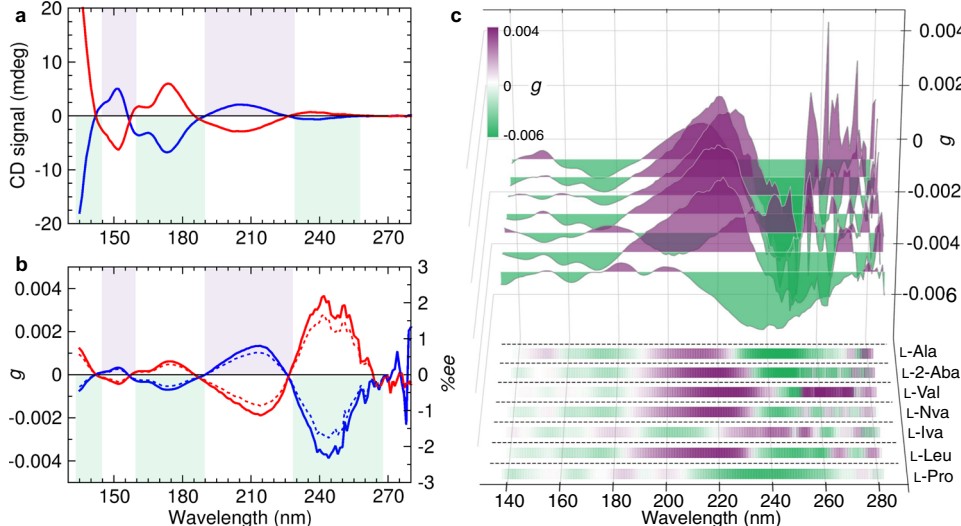

**Fig. 2 Chiroptical gas phase spectra of amino acid enantiomers. a** CD spectra of D-alanine (red); L-alanine (blue). Green and purple areas are indicative of the sign of the L-alanine CD spectrum for comparison with other amino acids. **b** Anisotropy (g) spectra of D-alanine (red) and L-alanine (blue) (thick lines) revealing the %enantiomeric excess (%ee) inducible by either left or right circularly polarized light at the extent of reaction ξ = 0.9999 (dotted lines). **c** Anisotropy spectra (g) of seven L-amino acids. Green/purple color and brightness indicate negative/positive and magnitude, respectively, of g values. Source data of CD and g spectra are provided for all amino acid enantiomers as a Source Data file.

**Table 1 Amino acid gas phase chiroptical properties and inducible ee values for an extent of reaction ξ = 0.9999.**

| Compound | Transition A | | | Transition B | | | Transition C | | |
|---|---|---|---|---|---|---|---|---|---|
| | λ/nm | g | ee (%)[a] | λ/nm | g | ee (%)[a] | λ/nm | g | ee (%)[a] |
| L-Alanine | 244 | −0.0039 | \|1.77\| | 214 | 0.0013 | \|0.61\| | 173 | −0.0007 | \|0.32\| |
| L-2-Aminobutyric acid | 244 | −0.0042 | \|1.90\| | 220 | 0.0029 | \|1.35\| | 176 | −0.0005 | \|0.25\| |
| L-Proline | 236 | −0.0021 | \|1.62\| | 200 | 0.0002 | \|0.20\| | 192 | −0.0001 | \|0.08\| |
| L-Valine | | | | 226 | 0.0031 | \|1.42\| | 176 | −0.0004 | \|0.17\| |
| L-Norvaline | 245 | −0.0016 | \|0.75\| | 219 | 0.0025 | \|1.14\| | 175 | −0.0006 | \|0.27\| |
| L-Isovaline | 246 | 0.0017 | \|0.77\| | 203 | −0.0008 | \|0.38\| | 179 | 0.0003 | \|0.12\| |
| L-Leucine | 247 | −0.0019 | \|0.88\| | 221 | 0.0027 | \|1.22\| | 174 | −0.0007 | \|0.30\| |

[a]The lower limit of inducible ee is calculated using[21]: $|\%ee| \geq (1 - (1-\xi)^{\frac{|g|}{2}}) \times 100\%$, with $\xi = 0.9999$.

The newly recorded CD and anisotropy spectra of gas phase amino acids greatly differ in band position and intensity from those recorded in solution[22,23] and their isotropic solid state[21,24]. Obviously, the available spectral region in solution is limited due to solvent absorption. The CD and anisotropy spectra in aqueous solution of all amino acids are dominated by a single broad band at about 205 ± 10 nm corresponding to the $n_O \rightarrow \pi^*_{CO}$ electronic transition of the carboxylate anion chromophore[25]. Whereas solid-state absorption and CD spectra of amino acids provided new insights into the intrinsic amino acid structures in the absence of any solvent and enabled chiroptical measurements in the far UV region[21,24], it still probes the amino acid zwitterion structure rather than the inherent properties of the amino acids' non-ionic forms as in the gas phase. Moreover, interactions of adjacent molecules in the isotropic condensed phase, leading to conformational changes and spectral congestion, are expected.

**Conformer-specific CD and band assignment from quantum chemistry.** Quantum chemical ab initio calculations were used to interpret the spectroscopic results. Amino acids are flexible molecules due to multiple torsional degrees of freedom. Their conformational landscape in the gas phase is strongly influenced by stabilization through intramolecular hydrogen bonding, which results in numerous low-energy conformers that contribute

independently to the overall electronic spectrum and challenges the prediction of the conformational distribution. Several basis sets have been selected for an in-depth analysis of gas-phase CD and anisotropy spectra of alanine and details are provided in the Supplementary Methods. The best agreement between theory and experiment for the excitation energies and transition moments on the time-dependent density functional theory (TDDFT) level was obtained applying the M06-2X/aug-cc-pVQZ basis set (Supplementary Fig. 12). High level structural data and relative energies were obtained on the explicitly correlated coupled cluster level (CCSD(T)-F12), with the corresponding CD signals being computed on the approximate second order coupled cluster level CC2 (Supplementary Figs. 13 and 14 and Supplementary Data 1 and 2).

Gas phase alanine reportedly has five energetically close-lying conformers in the ground electronic state, which are connected by low energy barriers[14,15,26,27]. To sample the most abundant conformers, we ran a typical flowchart combining molecular dynamics simulations and quantum calculation[28]. Seven stable conformations with relative energies of up to 8 kJ mol$^{-1}$ (Fig. 3a) were considered for TDDFT calculations and a Boltzmann distribution was applied to the set of conformers at different temperatures (300 and 460 K). Comparison of these conformers with previous work is reported in Supplementary Table 3. Temperatures in the range of 433–458 K were necessary to

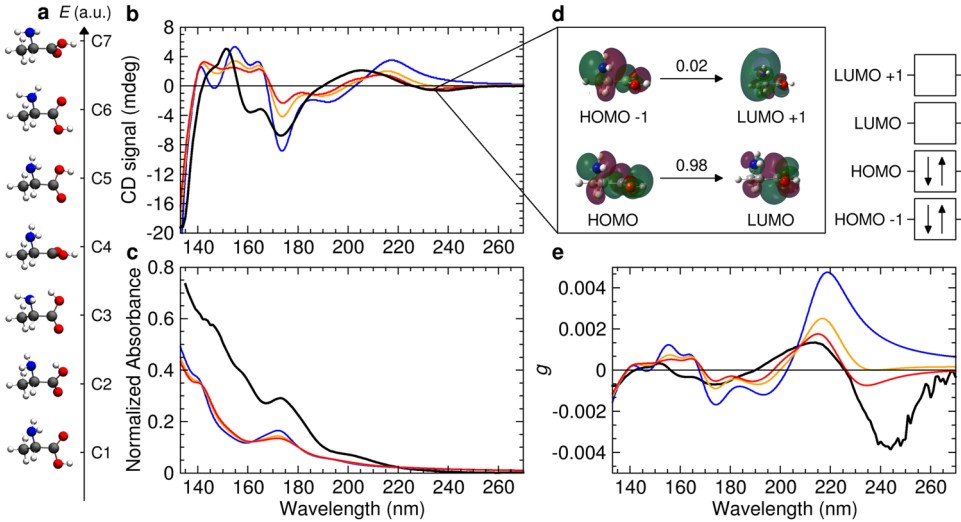

**Fig. 3 Time-dependent density functional theory derived chiroptical gas phase spectra of ʟ-alanine. a** The seven low-energy conformers considered for theoretical calculations (M06-2X/aug-cc-pVQZ). **b**, **c**, **e**, Comparison of experimental gas phase (black line) and theoretical spectra of ʟ-alanine (M06-2X/aug-cc-pVQZ) calculated at three different temperatures, $T = 0$ K, lower energy (blue), Boltzmann weighted at $T = 300$ K (yellow), and Boltzmann weighted at $T = 460$ K (red). **b** CD spectra, **c** absorbance spectra and **e** anisotropy spectra. **d** TD-DFT calculations allow for the attribution of the first chiroptical transition with the oscillator strengths $f$ given for the involved natural transition orbitals.

provide adequate vapor pressures for CD and anisotropy measurements. TDDFT calculations show that the energy and intensity of the chiroptical transitions do not vary significantly in between $T = 0$ K and $T = 460$ K, importantly suggesting that the experimental conditions are consistent with those expected for the interstellar medium. Differences between measured and DFT-calculated absorbance below 200 nm (Fig. 3c) is explained by both the presence of photodegradation products, such as water and ammonia in the gas cell (Supplementary Figs. 9 and 10), and due to deficiencies of the quantum chemical methods when approaching the ionization threshold. The photodegradation products (Supplementary Note 3 and Supplementary Fig. 11) were kept to a minimum by operating the gas cell under a flow condition, where the sample was constantly replenished via simultaneous inlet and pumping (Supplementary Methods). The experimental anisotropy values represent therefore a lower limit, especially in the far UV.

The long-wavelength band only appears when considering alanine conformers at higher temperature (Fig. 3b, e). The $g$ band observed at 244 nm, with comparatively broad vibronic profile characteristic of a $n_O \rightarrow \pi^*_{CO}$ (HOMO-LUMO) transition in carbonyl compounds (Fig. 3d), results predominantly from the higher energy conformer 4 (Supplementary Figs. 12 and 14). This conformer adopts an unusual conformation with the amine and carboxylic group found in two orthogonal planes (Fig. 3a), while it is also adopting a planar *cis* arrangement of the carboxyl functional group, which increases its stability[27]. Previous experimental work allowed the detection of a total of four individual neutral alanine conformers including conformers 1 and 2 by electron diffraction[29], conformers 1 and 3 by microwave[14,15] and PECD spectroscopy[16], and conformer 5 by jet-cooled Raman spectroscopy[30]. To the best of our knowledge, our gas phase chiroptical analyses, as conducted in combination with quantum chemical calculations, provide the first experimental evidence of conformer 4 of neutral alanine. Gas phase CD and anisotropy spectroscopy can therefore offer crucial information for elucidating conformation-specific configurations, as CD signs and magnitudes are very sensitive to conformations.

Whereas the CD spectrum at 460 K estimated on the TDDFT level remains nearly featureless in the 140–160 nm spectral region, both the CD spectrum on the coupled cluster level (Supplementary Fig. 13) as well as its experimental counterpart show pronounced structure. The latter displays inversion of the CD signal from positive to negative with a zero crossing near 157 nm, which is partially reproduced on the CC2 level, but found to be more expanded reaching down to 150 nm. This type of CD signal is typical for an exciton coupling between two close identical chromophores[31] and could therefore indicate the presence of a dipeptide. Investigations of polymerization and cyclization of alanine at elevated temperatures is found in the Supplementary Note 5 and shows that sublimation does not lead to the formation of significant amounts of dipeptides or diketopiperazines (Supplementary Fig. 15). We found that commercially available ʟ-alanine contains less than 1% of ʟ-alanyl-ʟ-alanine; hence, dipeptides are not a priori expected to influence the alanine spectrum significantly. However, since exciton bands are often more intense than CD bands from the monomer[32,33] even small amounts of dipeptides could potentially contribute to the difference from the calculations.

While we can confirm the assignments of previously proposed states of the electronic spectrum of neutral alanine[34], our experimental and theoretical data extend our understanding of hitherto unassigned bands of neutral amino acids. Most of the primary low-lying singlet transitions in gas phase amino acids can be attributed to the valence excitation of the highest lying non-bonding molecular orbital localized at the carboxylic oxygen atom ($n_O$). Thus, besides the broad $n_O \rightarrow \pi^*_{CO}$ band, a series of transitions belonging to the $n_O \rightarrow 3\,sp$ Rydberg excitations are predictable (Table S1). Moreover, our gas phase data allow for the measurement of the first Rydberg excitations from the non-bonding lone pair on the nitrogen atom ($n_N \rightarrow 3sp$) of the neutral amine group, inaccessible in solution and solid state. Theoretical approaches to assign the observed CD bands of higher lying excited states to specific orbital transitions become less accurate as significant mixing occurs. We tentatively assign the valence $\pi_{C=O} \rightarrow 3s$ and $\pi_{C=O} \rightarrow \pi^*_{CO}$ states of neutral amino acids to be located below 155 nm. Upon quantitative and qualitative examination, important differences between proline and all other amino acids come to the fore as the five-membered pyrrolidine ring introduces significant distortions to the conformational

space, so that band positions, signs, and magnitudes of all electronic excitations differ considerably.

## Discussion

**Gas phase optical activity as the origin of extra-terrestrial symmetry breaking.** Life's amino acid preference for left-handedness is still a central puzzle in modern biochemistry, with several theoretical and experimental causes discussed[35], refs therein such as molecular parity violation, selective adsorption on minerals surfaces, and enantioselective interactions with circularly polarized light or spin-polarized electrons. The detection of L-enriched amino acids in the Murchison meteorite[36,37] suggested the possibility of an extra-terrestrial contribution to the missing link between purely racemic molecules on prebiotic earth and homochirality of life. Substantial progress has been made in the understanding of the photochemical formation of life's molecular building blocks in interstellar molecular clouds based on the identification of amino acids[38], aldehydes[39], and sugar molecules[40] in simulated interstellar pre-cometary ices. Obtained results were confirmed by in situ data of ESA's cometary Rosetta mission and its COSAC[41] and Ptolemy[42] instruments on board the Philae Lander. Most of the current models of comet nuclei presume that to a major extent they are the aggregates of interstellar icy grains in their evolved state in the collapsing molecular cloud. Gas and solid-state chemical reactions induced by UV photolysis (unpolarized and polarized), cosmic rays or thermal cycling form and process the complex organic inventory of interstellar dust that is later integrated in the presolar nebula from which planets and small solar system bodies formed[43]. Besides, Rosetta's ROSINA instrument detected the achiral amino acid glycine in the cometary coma[44] allowing the assumption that comets may contain more complex amino acids that can desorb into the comet's atmosphere at sufficiently high temperatures.

As several sources of naturally polarized light have been detected in interstellar star-forming regions[45,46] and given the strong evidence that the solar system originated in a high-mass star-forming region[47], this has been considered to further strengthen the hypothesis of a photochemical origin of life's homochirality. Most relevant astronomical circularly polarized light (CPL) sources are reflection nebulae in high-mass star forming regions that show high degrees of circular polarization due to dichroic extinction of linearly polarized scattered light[45,46]. All CPL sources so far exhibit a quadrupolar pattern of left-handed and right-handed CPL which is explained by the scattering of the bipolar outflow lobes where the right-handed or left-handed CP regions are situated at the opposite sides of the outflow axis as well as at the opposite sides of the central illuminating source. Each quadrant of single-handed CPL extends on very large spatial scales of up to 0.65 pc[46], which is hundreds of times the size of most planetary-forming systems, including our solar system. Assuming our solar system formed in a similar high-mass star-forming region, all organic molecules in the gas phase or condensed on icy grains would have been illuminated by CPL of one helicity only during protoplanetary disk evolution[48,49].

Anisotropy spectroscopy of chiral molecules is a direct measure of the selectivity and efficiency of circularly polarized light to induce optical activity and enrich one enantiomer over the other via asymmetric photolysis[21]. Corresponding experiments to understand the transfer of asymmetry from chiral photons to racemic starting material demonstrated that CPL can induce asymmetric photochemistry in alanine[50] and proline[17], as well as being capable of asymmetric photochirogenesis of several amino acids in interstellar analog ices[48,49]. Due to an interstellar dust cycling process[43], by which organic molecules such as amino acids can reversibly and continuously desorb and condense back onto the surface of cometary and pre-cometary dust particles, chiral photon-molecule interactions are assumed to have—at least partly—taken place in the gas phase of interstellar environments. Such desorption–condensation cycles may have been crucial for an *ee* amplification via asymmetric photolysis of interstellar ices intimately connected with the surrounding gas, i.e., starting from $10^{-3}$ *g* values, as reported here and $10^{-3}$ to $10^{-2}$ *g* values in the solid state[21], to a few %*ee* as identified in carbonaceous chondrites. In such a scenario, the origin and evolution of biomolecular asymmetry would be considered as a two-step process, with an initial mirror-symmetry breaking event in the life cycle of interstellar gas and dust during cloud evolution followed by nonequilibrium reaction networks on the early Earth driving prebiotic molecular systems toward a homochiral state[51,52]. Our gas phase chiroptical data on neutral amino acids therefore complement previous CPL-based investigations on zwitterionic amino acids in the solid state[50] and the absolute asymmetric photosynthesis of amino acids in interstellar analog ices[48,49].

Stellar CPL, which is polychromatic, exhibits a dominant UV emission[45] matching the wavelength range of the anisotropy extrema of the investigated amino acids studied here. Given the decline in UV photon flux of most stars in the far UV range (<200 nm)[45,53], the broad anisotropy $n_O \rightarrow \pi^*_{CO}$ band (220–250 nm) would dictate the inducible *ee* in almost all gas phase amino acids, except valine, where the $n_N \rightarrow 3s$ band (190–220 nm) would dominate the interstellar asymmetric photolysis processes. Because stellar CPL would induce a net *ee* of the same handedness in almost all amino acids, such gas phase asymmetric photochemical reactions might have originally contributed to the evolution towards an enantiomeric selection of life's L-amino acids. The gas phase chiroptical and absorption spectra reported here are moreover crucial to trigger the UV-spectroscopic identification of chiral molecules in interstellar and circumstellar space, including their enantiomeric excess determination.

## Methods

**Gas-phase circular dichroism and anisotropy measurements.** Circular dichroism spectra CD($\lambda$), anisotropy spectra *g*($\lambda$), and absorption spectra A($\lambda$) of seven pairs of amino acid enantiomers were recorded using a temperature and pressure-controlled gas cell at the ASTRID2 synchrotron radiation source, Aarhus University, Denmark (Supplementary Methods). The cell consists of a stainless-steel tube, 500 mm in length with an 18 mm inner diameter, mounted on the AU-CD beam line. The pressure inside the gas cell is monitored using a type 631D Baratron heated manometer (MKS instruments), while the cell is evacuated using a Varian V70 turbo pump. Two all-metal valves are mounted on the main gas cell tube: one connects, via a flexible hose, to the turbo pump, while the other valve connects to the heated sample reservoir. Heating tapes and cartridge heaters were used to heat the entire gas cell up to the required temperature using PID computer control, with a maximum possible temperature of 200 °C. Differential heating of the windows to 5–10 °C higher temperatures compared to the tube avoids local condensation of analytes.

Typically, less than 1 g of a given amino acid enantiomer was placed into a glass container (Ø12 mm, length 15 mm) and inserted into the sample reservoir heater. Absorbance and CD spectra of each amino acid were recorded starting at $T = 160$ °C with gradually increased temperatures until sufficient vapor pressures with optimal optical densities were reached. The gas cell was typically operated under flow conditions, where the valves to the sample and the turbo pump were partially open, to ensure a continuous renewal of the gas in the cell.

**Gas chromatographic measurements.** The GC × GC–TOF-MS Pegasus IV D system from operated at a storage rate of 100 Hz, with a 50–400 amu mass range and a detector voltage of 1.45 kV (Supplementary Methods). Data were processed using the LECO Corp. ChromaTOF™ software. The source and transfer temperature were kept at 230 and 240 °C, respectively. The column set consisted of a Chirasil-L-Val column (25 m × 0.25 mm inner diameter, 0.12 mm film thickness) in the first-dimension modulator-coupled to a DB Wax secondary column (1.5 m × 0.1 mm inner diameter, 0.1 mm film thickness). Helium was used as carrier gas at a constant flow of $\bar{u} = 1$ mL min⁻¹. Sample volumes (1 μL) were injected in splitless mode at an injector temperature of 230 °C. The GC primary oven was operated as follows: 40 °C (1 min), warm up to 80 °C (10 min) at 10 °C min⁻¹, and warm up to 190 °C (15 min) at 4 °C min⁻¹. The secondary oven used the same temperature program with a constant temperature offset of 20 °C. A modulation period of $P_M = 5$ s was applied.

**Liquid chromatography measurements**. The HPLC Agilent 1200 system was composed of a quaternary pump (Agilent G1311 A) equipped with a Diode Array Detector (DAD Agilent G1315D) and an ELSD (Evaporative Light Scattering Detector). A Luna C18 column (Phenomenex, 150 × 4.6 mm, 5 μm) was used at 25 °C with an injection volume set at 10 μL and a flow rate set at 1.0 mL min$^{-1}$. The HPLC was used in isocratic mode with 95% chromatography grade water (A) (acidified with 1% formic acid) and 5% methanol (B) during 40 min.

**Computational details**. A thorough conformational search was achieved by a biased molecular dynamics simulation following an Umbrella Sampling protocol. A single alanine residue was described in the neutral state using the ff14SB force field. All 22 selected alanine structures were refined with Gaussian16 using a combination of different methods and a large basis set quadruple-$\zeta$ with polarization and diffuse functions: aug-cc-pvQZ. A single-point energy calculation with zero-point corrected energies including frequency calculations was done to verify their nature of true minima. Three methods were evaluated: CAM-B3LYP, ωB97X-D, and M06-2X. Finally, seven low energy conformers were further considered for calculations. TD-DFT calculations were done on the seven optimized structures using the three previous functionals combined with the aug-cc-pvQZ basis set. Two hundred excited states were calculated for each optimized conformer.

For the coupled cluster calculations, Csaszar's[27] study of alanine conformers was used as a starting point, from which all reported 13 local minimum structures were obtained and subsequently Boltzmann-weighted. A total of nine alanine conformers were included in the computation of the CD spectrum. All 13 conformer structures were energy optimized with the program package Molpro on the df-CCSD(T)-F12 level with the aug-cc-pVDZ-F12 basis set, specifically with the F12b method and the 3*C(FIX, HY1) ansatz as implemented in Molpro and their cartesian coordinates in Å are provided in the Supplementary Data 1. Harmonic vibrational wavenumbers were calculated on the same level (Supplementary Data 2). Relative energies were determined by single point calculation with CCSD(T)-F12 and aug-cc-pVTZ-F12 basis set on the structures optimized with the double zeta basis set. Electronic excitation energies and intensities of the CD and the one-photon absorption spectra were calculated with the program package Turbomole with the RI-CC2 method and the aug-cc-pVQZ basis set.

## Data availability
The authors declare that all data supporting this study are available within the main text and Supplementary Information file. A detailed experimental material and method section as well as Supplementary Figs. 1–15 and Tables 1–3 can be found in the Supplementary Information file. Raw data of the individual circular dichroism and anisotropy data are provided in the Source data file. Source data are provided with this paper.

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

## Acknowledgements

This work was funded by the European Research Council under the European Union's Horizon 2020 research and innovation program (grant agreement 804144, C.M.). Further funding was provided by the French government through the UCA^JEDI Investments in the Future project managed by the National Research Agency (ANR), with reference number ANR-15-IDEX-01 (C.M.), and by the ANR under grant number ANR-18-CE29-0004-01 (U.J.M.), as well as by the project CALIPSOplus, under Grant Agreement 730872 (S.V.H., N.C.J., and C.M.) from the EU Framework Program for Research and Innovation HORIZON 2020. Financial support from the Deutsche Forschungsgemeinschaft (DFG, German Research Foundation)—Project number 328961117—SFB 1319 ELCH (M.D. and R.B.) is acknowledged. A.D.G. is grateful for a PhD scholarship from the French Ministry of Science and Education. Computation for the work described in this paper was supported by the Université Côte d'Azur's Center for High-Performance Computing. We thank Dr. Jana Bocková for supporting the alanine sublimation experiments as well as Dr. Thomas Michel for the liquid chromatographic analyses.

## Author contributions

U.J.M., C.M., L.N. and S.V.H. conceived and designed the experiments. C.M., A.D.G., J.T., N.C.J. and S.V.H. performed the spectroscopic investigations, J.T., M.D. and R.B. the calculations. C.M. assembled and wrote the Supplementary Information with contributions from J.T., N.C.J. and S.V.H. A.D.G. and J.T. contributed equally to the experiments and are listed alphabetically. C.M. wrote the manuscript with the contribution of all authors.

## Competing interests

The authors declare no competing interests.
