## [Peer Review File · Nature Communications]

REVIEWER COMMENTS

Reviewer #1 (Remarks to the Author):

Meinert and co-workers report the gas-phase CD spectra of neutral amino acids. Obtaining these is a complicated experimental endeavour and the computational analysis reads sound. There are just two comments from my side:

1) "The g band observed at 244 nm, with comparatively broad vibronic profile characteristic of a $nO \rightarrow \pi^*CO$ (HOMO-LUMO) transition in carbonyl compounds (Fig. 2d), results predominantly from the higher energy conformer 4"

The reader should be directed to Fig S11 here in order to support this statement. In fact, when reading it for the first time, my thought was: "What if they have just simulated the spectrum with a too large bandwidth and the negative feature of other conformers disappears". With the line spectra in S11 it is made clear that conformer 4 is really the only one that shows this band.

2) Did the authors try to measure also D-amino acids to (a) confirm the validity of the experimental spectra and (b) to show that the opposite enantiomer really shows the opposite signed band that confirms the presence of conformer 4?

Given that the experiment is far from trivial and that the experimental results can serve as a benchmark also for the development of high-accuracy CD calculations, the manuscript certainly deserves publication. In light of this benchmarking character, I would strongly urge the authors to provide the experimental spectra as csv-files as part of the SI, so that they can easily be retrieved by others.

Reviewer #2 (Remarks to the Author):

The work of C. Meinert et al. reports on CD and anisotropy spectra of amino acids recorded in the gas phase. Their experiments are complemented by quantum chemical calculations.

The analysis of their results is then used to support what could be called the "asymmetric photochemistry" route towards unraveling the mystery behind Nature's biomolecular asymmetry (i.e., the exclusive presence of chiral amino acids

in their stereochemical L-form).

This is overall a very nice work. For example, thanks to the authors, we have the first experimental data on CD and anisotropy spectra for enantiomeric pairs of gas phase amino acids including alanine, 2-aminobutyric acid, proline, valine, norvaline, isovaline, and leucine in the vacuum-ultraviolet and far UV regions.

At the same time, the quantum chemical calculations reported use a sufficiently accurate level of theory to draw conclusions that are used to rationalize the experimental results. In particular,

the authors discuss the appearance of correct features in the coupled-cluster (CC2) spectra vs. their absence in the corresponding TDDFT simulations.

It should be nonetheless pointed out that this type of calculations on such relatively small molecules are quite standard and highly automated nowadays, hence they do not represent a challenging task overall for the presented work of research.

In short, I have basically no criticisms against the presented results, but I am rather conservative with respect to the main claim of the paper, namely that there is enough evidence to promote a photochemistry-based explanation of the origin of the observed L-stereochemistry in biological systems.

In order to support their statement, the authors should provide comparison with other proposed explanations. At most, what their results show is that circularly polarized light is a possible candidate driving force - among many - that could have produced an enrichment of one enantiomer over the other.

But for example, why would such mechanism lead to a complete absence of the D-stereochemistry?

For the above reasons, and regretfully, I would not consider the present paper suitable for publication as Nature Communication. The results are nonetheless of interest to that segment of

the scientific community interested in spectroscopy, and thus I would suggest publication in any journal that is addressed more specifically to that community instead.

Reviewer #3 (Remarks to the Author):

Meinert et al. constructed new tool to observe the chiroptical properties of gas phase amino acids for exhibiting the study of the origin of biomolecular asymmetry. The research achievements would be significance for the related fields and the first crucial step to confirm the presence of molecular symmetry breaking under the gas phase, which would be possible condition of organic molecules in the interstellar environments. Further, the measurement system of the circular dichroism spectra is described in detail and the measurements are carefully conducted monitoring the temperature and gas pressure. I would like to put some comments and questions for the improvements of the paper and the deeper understandings.

The solid state of enantiomer has still large potential on the molecular symmetry breaking in the interstellar environments as your group have reported so far. In this paper, you switched the environments to gas from solid states. Are there any defects or problems on the studies of solid states of enantiomers? It would be better to describe the differences and similarities between the studies of gas and solid states.

The accuracy or amount of error of the chiroptical gas phase spectra of amino acid enantiomers are unclear. I think that the spectral intensity would differ depending on the gas pressure, ideally keeping the spectral shapes. Did the spectra show the similar spectral shapes for the different gas pressure? If different shapes, could you give any error bar in the spectra?

Line 10, page 2: Authors mentioned the chiroptical measurements in the gas phase had so far been inaccessible for amino acids because of insufficient vapor pressure of the target molecules. Are there any significant improvements in the chiroptical measurement system in the gas phase in your study compared to the past literatures?

Line 23, page 4: Authors calculated the theoretical spectra using several basis sets and compared them with the experimental ones. It is unclear how you decided the best agreement from S9. Did you estimate the spectral differences between experiment and theory, for examples, using the root mean square deviation? Further, are there any reasons that the half-width is 0.4 eV?

Line 18, page 5: Your group used synchrotron radiation for the measurements but the photodegradation products such as water and ammonia have large absorbance in the far UV region, giving a low wavelength limitation in the experimental anisotropy values. These products would be inevitable effect but if we can estimate the partial pressure of water and amino acids using other experimental method or theory, is it possible to remove such lower limit?

We provide here point-by-point responses to the reports of reviewer #1 to #3. The reviewers' reports are copied below in *italics* and our responses are inserted in plain text. Changes in the manuscript are shown in the manuscript and supplementary text file with track changes. Moreover, we have added the source data of the spectra within a zipped folder.

Reviewer #1 (Remarks to the Author):

Meinert and co-workers report the gas-phase CD spectra of neutral amino acids. Obtaining these is a complicated experimental endeavour and the computational analysis reads sound. There are just two comments from my side:

*(1) "The g band observed at 244 nm, with comparatively broad vibronic profile characteristic of a $n_o \rightarrow \pi^*_{CO}$ (HOMO-LUMO) transition in carbonyl compounds (Fig. 2d), results predominantly from the higher energy conformer 4". The reader should be directed to Fig S11 here in order to support this statement. In fact, when reading it for the first time, my thought was: "What if they have just simulated the spectrum with a too larger bandwidth and the negative feature of other conformers disappears". With the line spectra in S11 it is made clear that conformer 4 is the really the only one that shows this band.*

Response #1-1: We are grateful for the referee's advice and refer now in the text to the Supplementary Figs. 10 and 12. The referee is right, the line spectra of Fig. S12 (formerly Fig. S11) shows the contribution of the individual conformers to the overall spectra of L-alanine with conformer 4 (III B) being the only conformer exhibiting a negative transition at long wavelengths. Moreover, we describe in the Supplementary text S7 "if conformer 4 is removed from the Boltzmann weighted CD spectra at $T = 460$ K using the hybrid functional M06-2X, the lowest lying CD band disappears completely indicating that conformer 4 dominates the CD sign and magnitude of the $n_o \rightarrow \pi^*_{CO}$ transition (**Supplementary Fig. 10e, f**)."

(2) Did the authors try to measure also D-amino acids to (a) confirm the validity of the experimental spectra and (b) to show that the opposite enantiomer really shows the opposite signed band that confirms the presence of conformer 4?

Response #1-2: Yes, we did record the CD and anisotropy spectra of both enantiomers of all presented amino acids including alanine, 2-Aba, proline, valine, norvaline, isovaline, and leucine. The full set of experimental data (L- and D-enantiomers) are shown in the originally submitted Supplementary Files (S5, Figs S2-S8). The referee is right that the measurements of both enantiomers allow for evaluating the quality and validity of the experimental spectra. All enantiomeric pairs showed excellent mirroring effects, except D-/ L-proline for which the L-enantiomer showed an overall lower intensity, most probably due to lower enantiopurity (see Supplementary Figure 4).

The mirroring effect in the CD and anisotropy spectra of L- and D-enantiomers of alanine (but also of all other measured amino acids) confirms our experimental data, and in combination with our theoretical investigations, the presence of alanine's conformer 4.

We believe that we have sufficiently highlighted the measurement of both enantiomers at the following instances in our manuscript:

*page 2/line 18-19: “we have now recorded the first CD and anisotropy spectra for **enantiomeric pairs** of gas phase amino acids including....”

*page 2/line 30-32: “The **L-alanine enantiomer** shows negative CD states at **D-alanine shows the expected precise mirroring effect (Fig 1).**”

Given that the experiment is far from trivial and that the experimental results can serve as benchmark also for the development of high-accuracy CD calculations, the manuscript certainly deserves publication.

(3) *In light of this benchmarking character, I would strongly urge the authors to provide the experimental spectra as csv-files as part of the SI, so that they can easily be retrieved by others.*

Response #1-3:

Thank you very much for your positive comment and suggestion. We do agree with you that the csv-files or .text files will be a great addition to the Supplementary files. We prepared the final experimental data set for both enantiomers (L- and D-) of each amino as a separate csv file that can easily be downloaded by experimentalists and theoreticians interested in our experimental results.

Reviewer #2 (Remarks to the Author):

The work of C. Meinert et al. reports on CD and anisotropy spectra of amino acids recorded in the gas phase. Their experiments are complemented by quantum chemical calculations. The analysis of their results is then used to support what could be called the "asymmetric photochemistry" route towards unraveling the mystery behind Nature's biomolecular asymmetry (i.e., the exclusive presence of chiral amino acids in their stereochemical L-form).

This is overall a very nice work. For example, thanks to the authors, we have the first experimental data on CD and anisotropy spectra for enantiomeric pairs of gas phase amino acids including alanine, 2-aminobutyric acid, proline, valine, norvaline, isovaline, and leucine in the vacuum-ultraviolet and far UV regions. At the same time, the quantum chemical calculations reported use a sufficiently accurate level of theory to draw conclusions that are used to rationalize the experimental results. In particular, the authors discuss the appearance of correct features in the coupled-cluster (CC2) spectra vs. their absence in the corresponding TDDFT simulations.

(1) *It should be nonetheless pointed out that this type of calculations on such relatively small molecules are quite standard and highly automated nowadays, hence they do not represent a challenging task overall for the presented work of research.*

Response #2-1:

We partly agree with this comment of referee #2. Indeed, these calculations are to some extent standard, if one judges only by the criterion that they were not done with a custom-made program. However, the explicitly correlated CCSD(T)-F12 calculations even for such small molecules are not only demanding, but also crucial to obtain definite results for equilibrium structures and relative energies to assemble the CD-spectra subsequently. Moreover, they provide benchmark data for DFT methods to determine functionals that are applicable to much larger systems. By our combined experimental and theoretical efforts, conformational details could be elucidated. Moreover, the theoretical determinations revealed advantages and limitations of each functionals used in the TDDFT simulations

as well as on the coupled-cluster (CC2) level to accurately account for all spectral features in the CD and anisotropy spectra of such *small molecules*. Implementing only one state-of-the-art computational approach in our workflow would have therefore been insufficient to reveal the presence of conformer 4.

(2) In short, I have basically no criticisms against the presented results, but I am rather conservative with respect to the main claim of the paper, namely that there is enough evidence to promote a photochemistry-based explanation of the origin of the observed L-stereochemistry in biological systems. In order to support their statement, the authors should provide comparison with other proposed explanations. At most, what their results show is that circularly polarized light is a possible candidate driving force - among many - that could have produced an enrichment of one enantiomer over the other. But for example, why would such mechanism lead to a complete absence of the D-stereochemistry?

Response #2-2:

We are thankful for the honest critics of referee #2 regarding the plausibility of a photochemical origin of life's biomolecular asymmetry. Indeed several theoretical^[1-3] and experimental^[4-16] proposals for chiral symmetry breaking inherent to all biopolymers have been considered. However, there are limitations in that either the chemistry involved is not prebiotically plausible (as in the Soai autocatalytic reaction^[15]) or they apply only to specific chiral molecules (as in chiral conglomerate crystals^[16]). Moreover, asymmetric autocatalytic systems, largely used to explain how mixtures yield products of high optical purity (homochiral state), do not promote mirror-symmetry breaking, but simply amplify a small initial chiral bias. It is crucial to note that, all chiral amplification models reported so far give rise to a random outcome in handedness in the absence of an external chiral influence.

Given the limitation in manuscript length and number of references, we added a new reference to the main text '*Amino acids and the asymmetry of life*' [ref 35]. This book provides the reader with a broad overview on various proposals for the original symmetry breaking event including the most popular hypotheses such as symmetry breaking by crystallization, selective adsorption of enantiomers on mineral surfaces, molecular parity violation, and enantioselective chemistries induced by circularly polarized light, or spin-polarized electrons. Please see our changes here:

Main text, Discussion (page 7):

Life's amino acid preference for left-handedness is still a central puzzle in modern biochemistry, with several theoretical and experimental causes discussed^{35 and refs therein} such as molecular parity violation, selective adsorption on minerals surfaces, and enantioselective interactions with circularly polarized light or spin-polarized electrons.

While we agree with referee #2 that there are several reasonable theories on the original cause of biomolecular homochirality, our motivation and current research efforts are driven by the enigmatic detection of L-enriched amino acids in multiple meteoritic samples and the understanding of such symmetry breaking in extra-terrestrial environments. Rather than claiming that the proposal of a photochemistry-induced L-stereoselectivity is superior compared with other proposed mechanism for the initial symmetry breaking effect, we focus on exploring a more realistic scenario of complex interstellar ice chemistries with implications of gas-dust cycles on the evolution of interstellar chiral molecules in non-equimolar ratios. Support and motivation for this idea are highlighted in the main

text, e.g. detection of L-enriched meteoritic amino acids, detection of several CPL sources in star-forming regions, detection of glycine in a cometary coma, laboratory simulations of photochemically produced interstellar ices using unpolarized and polarized radiation sources. To tone down the impact of our gas phase chiroptical property measurements on revealing the origin of life's homochirality, we have modified the title of our manuscript to the following:

Title: Amino acid gas phase circular dichroism and the origin of biomolecular asymmetry

Moreover, we have slightly rephrased the last paragraph to highlight a possible contribution of gas phase CPL 'only' and to leave the possibility of other mechanisms for the original selection of the biological enantiomeric form.

Main text, Discussion (page 9):

Because stellar CPL would induce a net *ee* of the same handedness in almost all amino acids, such gas phase asymmetric photochemical reactions might have originally contributed to the evolution towards an enantiomeric selection of life's L-amino acids.

Regarding the referee's comment on the complete absence of the D-stereochemistry, we realized that this has not been carefully enough addressed in the main text. We agree with the referee that a CPL-induced symmetry breaking is not leading to biomolecular homochirality *per se* but may have rather enriched extra-terrestrial material with *left-handed* amino acids (as seen in various carbonaceous chondrites). We therefore understand the origin and evolution of biomolecular asymmetry as a two-step process with an initial mirror-symmetry breaking leading to small imbalances in the amino acid enantiomer distribution followed by amplification models of mutualism and cooperative effects in autocatalytic sets that may have very well occurred on the early earth.

We have added the following statement to the discussion and two more references:

Main text, Discussion (page 8):

In such a scenario, the origin and evolution of biomolecular asymmetry would be considered as a two-step process, with an initial mirror-symmetry breaking event in the life cycle of interstellar gas and dust during cloud evolution followed by nonequilibrium reaction networks on the early Earth driving prebiotic molecular systems toward a homochiral state^{51,52}. Our gas phase chiroptical data on neutral amino acids therefore complement previous investigations on zwitterionic amino acids in the solid state⁵⁰ and the absolute asymmetric photosynthesis of amino acids in interstellar analogue ices^{48,49}.

Moreover, we have added the following paragraph to better explain the astrophysical context of the described CPL scenario:

Main text, Discussion (page 8):

[...] and given the strong evidence that the solar system originated in a high-mass star-forming region⁴⁷, [...]. Most relevant astronomical circularly polarized light (CPL) sources are reflection nebulae in high-mass star forming regions that show high degrees of circular polarization due to dichroic extinction of linearly polarized scattered light^{45,46}. All CPL sources so far exhibit a quadrupolar pattern of left- and right-handed CPL which is explained by the scattering of the bipolar outflow lobes where the right- or left-handed CP regions are situated at the opposite sides of the outflow axis as well as at

the opposite sides of the central illuminating source. Each quadrant of single-handed CPL extends on very large spatial scales of up to 0.65 pc⁴⁶ and refs therein, which is hundreds of times the size of most planetary-forming systems, including our solar system. Assuming our solar system formed in a similar high-mass star-forming region, all gas phase and condensed organic molecules would have been illuminated by CPL of one helicity only during protoplanetary disk evolution^{48,49}.

Refs

35. Meierhenrich, U. J. *Amino Acids and the Asymmetry of Life* (Springer, 2008).
47. Forbes, J. C., Alves, J. & Lin, D. N. C. A Solar System formation analogue in the Ophiuchus star-forming complex. *Nat. Astron.* **5**, 1009–1016 (2021).
48. De Marcellus, P. *et al.* Non-racemic amino acid production by UV irradiation of achiral interstellar ice analogs with circularly polarized light. *Astrophys. J. Lett.* **727**, L27 (2011).
51. Frank, F. C. On spontaneous asymmetric synthesis. *Biochim. Biophys. Acta* **11**, 459–463 (1953).
52. Laurent, G, Lacoste, D, Gaspard, P. Emergence of homochirality in large molecular systems. *Proc. Natl. Acad. Sci. U.S.A.* **118**, e2012741118 (2021).

For the above reasons, and regretfully, I would not consider the present paper suitable for publication as Nature Communication. The results are nonetheless of interest to that segment of the scientific community interested in spectroscopy, and thus I would suggest publication in any journal that is addressed more specifically to that community instead.

¹ Frank F. C. On spontaneous asymmetric synthesis. *Biochim. Biophys. Acta* **11**, 459–463 (1953).

² Yamagata Y. A hypothesis for the asymmetric appearance of biomolecules on earth. *J. Theoret. Biol.* **11**, 495–498 (1966).

³ Tranter G. E. Parity-violating energy differences of chiral minerals and the origin of biomolecular chirality. *Nature* **318**, 172–173 (1985).

⁴ Kondepudi D. K. Kaufman R. J. & Singh N. Chiral symmetry breaking in sodium chlorate crystallization. *Science* **250**, 975–976 (1990).

⁵ Soai K. *et al.* *d*- and *l*-Quartz-promoted highly enantioselective synthesis of a chiral compound. *J. Am. Chem. Soc.* **121**, 11235–11236 (1999).

⁶ Hazen R. M., Filley T. R. & Goodfried G. A. Selective adsorption of L- and D-amino acids on calcite: Implications for biochemical homochirality. *Proc. Natl. Acad. Sci. USA* **98**, 5487–5490 (2001).

⁷ Klussmann M., Izumi T., White A. J. P., Armstrong A. & Blackmond D. G. Emergence of solution-phase homochirality via crystal engineering of amino acids. *J. Am. Chem. Soc.* **129**, 7657–7660 (2007).

⁸ Cintas P. Sublime arguments: Rethinking the generation of homochirality under prebiotic conditions. *Angew. Chem. Int. Ed.* **47**, 2918–2920 (2008).

⁹ Ribó J. M., Crussats J., Sagués F., Claret J. & Rubires R. Chiral sign induction by vortices during the formation of mesophases in stirred solutions. *Science* **292**, 2063–2066 (2001).

¹⁰ Vester F., Ulbricht T. L. V. & Krauch H. Optische Aktivität und die Paritätsverletzung im β -Zerfall. *Naturwissenschaften* **46**, 66–68 (1959).

¹¹ Quack M. How important is parity violation for molecular and biomolecular chirality? *Angew. Chem. Int. Ed.* **41**, 4618–4630 (2002).

¹² Rosenberg R. A. Abu Haija M. & Ryan P. Chiral-selective chemistry induced by spin-polarized secondary electrons from a magnetic substrate. *J. Phys. Rev. Lett.* **101**, 178301 (2008).

¹³ Rikken G. L. J. A. & Raupach E. Enantioselective magnetochiral photochemistry. *Nature* **405**, 932–935 (2000).

¹⁴ Balavoine G., Moradpour A. & Kagan H. B. Preparation of chiral compounds with high optical purity by irradiation with circularly polarized light, a model reaction for the prebiotic generation of optical activity. *J. Am. Chem. Soc.* **96**, 5152–5158 (1974).

¹⁵ Soai K., Shibata T., Morioka H. & Choji, K. Asymmetric autocatalysis and amplification of enantiomeric excess of a chiral molecule. *Nature* **378**, 767–768 (1995).

¹⁶ Viedma C., *et al.* Evolution of solid-phase homochirality for a proteinogenic amino acid. *J. Am. Chem. Soc.* **130**, 15274–15275 (2008).

Reviewer #3 (Remarks to the Author):

Meinert et al. constructed new tool to observe the chiroptical properties of gas phase amino acids for exhibiting the study of the origin of biomolecular asymmetry. The research achievements would be significance for the related fields and the first crucial step to confirm the presence of molecular symmetry breaking under the gas phase, which would be possible condition of organic molecules in the interstellar environments. Further, the measurement system of the circular dichroism spectra is described in detail and the measurements are carefully conducted monitoring the temperature and gas pressure. I would like to put some comments and questions for the improvements of the paper and the deeper understandings.

(1) *The solid state of enantiomer has still large potential on the molecular symmetry breaking in the interstellar environments as your group have reported so far. In this paper, you switched the environments to gas from solid states. Are there any defects or problems on the studies of solid states of enantiomers? It would be better to describe the differences and similarities between the studies of gas and solid states.*

Response #3-1:

We do agree with the referee that the interaction of chiral photons with amino acids on interstellar water-rich dust grains should not be neglected. Indeed, most of the current models of comet nuclei presume that to a major extent they are basically aggregates of the interstellar dust in its final evolved state in the collapsing molecular cloud which becomes the protosolar nebula. In addition to the chemical consequences of such a model, there is the prediction of a morphological structure in which the aggregate material consists of tenth micron basic units each of which contains (on average) a silicate core, a layer of complex organic material, and an outer layer of ices in which are embedded all the very small carbonaceous particles (e.g. ^[1]). All these components have been observed in the comet comae in one way or another including gas phase glycine in the cometary comae of 67P/Churyumov-Gerasimenko by the Rosetta's ROSINA mass spectrometer^[2]. The detection of glycine has been a motivation of our consortium to study the chiral photon interaction of amino acids in the gas phase. We do believe, however, in the strong interplay between interstellar gas and dust and our gas phase chiroptical measurements of neutral amino acids should be understood as being complementary to the previously investigated zwitterionic amino acids in the solid state by our research group.

We believe that we have sufficiently discussed the fundamental differences of studying gas vs solid state amino acids. See for example: Main text, page 4/lines 6-12 where we highlight the neutral vs zwitterionic form of amino acids as well as expected conformational changes due to intermolecular interactions of adjacent amino acid molecules in the amorphous films absent in gas phase measurements. Nevertheless, the potential occurrence of asymmetric photochemical reactions of amino acids in the gas phase do not rule out asymmetric photochemical interactions in the condensed phase such as previously proposed by our team as well as several other research teams (e.g. Nishino *et al.* 2013, *Chem Eur. J.*, 19; Takano *et al.* 2007, *EPSL*, 254; Kaneko *et al.* 2009, *J. Phys. Soc. Jpn.* 78; Nakagawa *et al.* 2005, *J. Electr. Spectros.* 144; Takahashi *et al.* 2009, *Int. J. Mol. Sci.*, 10; Tanaka *et al.* 2009, *J. Synchr. Rad.*, 16; Tanaka *et al.* 2010, *J. Phys. Chem. A*, 114. etc.). Our newly experimental results on gas phase amino acids as neutral molecules complement our knowledge on amino acids as zwitterions in the solid and liquid state but do not rule each other out regarding their potential contribution to asymmetrical photochemistry in interstellar/circumstellar environments.

To best address the referee's comment on gas vs solid state amino acids, we have edited the main text as follows:

Main text, discussion (page 8):

Such desorption-condensation cycles may have been crucial for an *ee* amplification *via* asymmetric photolysis of interstellar ices intimately connected with the surrounding gas, i.e., starting from 10^{-3} *g* values, as reported here and 10^{-3} to 10^{-2} *g* values in the solid state²¹, to a few %*ee* as identified in carbonaceous chondrites. We therefore understand the origin and evolution of biomolecular asymmetry as a two-step process, with an initial mirror-symmetry breaking event in the life cycle of interstellar gas and dust during cloud evolution followed by nonequilibrium reaction networks on the early Earth driving prebiotic molecular systems toward a homochiral state^{49,50}. Our gas phase chiroptical data on neutral amino acids therefore complement previous investigations on zwitterionic amino acids in the solid state⁴⁷ and the absolute asymmetric photosynthesis of amino acids in interstellar analogue ices⁴⁸.

¹ Altwegg, K. *et al.* Prebiotic chemicals – amino acid and phosphorus – in the coma of comet 67P/Churyumov-Gerasimenko. *Sci. Adv.* **2**, e1600285 (2016).

² Potapov, A, Bouwman, J, Jaeger, C & Henning, T. Dust/ice mixing in cold regions and solid-state water in the diffuse interstellar medium. *Nature Astronomy* **5**, 78–85 (2020).

(2) *The accuracy or amount of error of the chiroptical gas phase spectra of amino acid enantiomers are unclear. I think that the spectral intensity would differ depending on the gas pressure, ideally keeping the spectral shapes. Did the spectra show the similar spectral shapes for the different gas pressure? If different shapes, could you give any error bar in the spectra?*

Response #3-2:

The referee is right. The magnitude of the spectra differs depending on the gas pressure. In the Supplementary text S5 we already described the fact that the gas cell pressure was not a direct measure of the amino acid gas density due to other (achiral) contributions, e.g. water. Accuracy and reproducibility of our data were achieved measuring several CD spectra for each amino acid enantiomer varying the cell temperature and flow rates hence different amino acid gas densities. Best spectra were scaled using a factor derived from the absorbance spectra in the wavelength range 170–230 nm (see details in S5). This procedure yielded almost perfect mirrored CD spectra for L- and D-alanine confirming the accuracy of our gas phase CD measurements and therefore has been applied to all other enantiomeric pairs of measured amino acids. We also like to stress out that the normalization had no impact on the anisotropy spectra as they are independent of such scaling.

To best address the referee's comment on the potential difference in spectral shape for different gas pressures, we have added a new chapter S6 on '*Shape dependence of CD spectra on gas pressure*' to the Supplementary file and refer in the main text to chapter S6 as well. In summary, no spectral changes have been observed within the range of gas pressures used to record the CD spectra. We provide additional details using multiple independent CD and absorbance measurements of alanine (newly added Fig S9) to underline the reproducibility and overall robustness of our experiments. Some of the recorded spectra of alanine showed higher contribution of ammonia, however, the spectral shape is maintained over a fairly large range of gas pressures used which becomes clear when looking at the normalized data:

Supplementary File, section S6 (page 11):

The CD and corresponding absorbance spectra of both the D- and L-enantiomer of alanine measured at various gas pressures are shown in **Supplementary Fig. 9a** and **9b**, respectively. Although the signal strengths obviously depend on the gas pressure, there is no apparent shape dependence on the pressure. To further elucidate if there is any shape change with sample pressure, the absorbance spectra have been scaled to the same values in the wavelength range 170–190 nm (**Supplementary Fig. 9d**), and the corresponding scaling factors are applied to the CD spectra as shown in **Supplementary Fig. 9c**. The scaling of the absorbance spectra merges all individual spectra together for wavelengths above 160 nm. The variation observed below 160 nm is attributed to the lower level of light intensity combined with the higher absorbance in this wavelength range, as well as to some extent from the nitrogen purge gas that is present in the beam path outside the gas cell (S2). Performing the scaling to the CD spectra brings them close to each other in a small range of overall signal magnitudes, but also highlights that no shape change is visible in the collection of a total of 15 different spectra. Therefore, within the range of gas pressures obtainable in the experimental set-up, all below 0.1 mbar, we do not observe any gas pressure induced shape change in the CD spectra. Within the various datasets presented below, there are some spectra which upon close inspection show a contribution from ammonia in the absorbance spectrum which will affect the results of the normalisation of the CD spectra and hence the anisotropy spectra (see S8 for more discussion on ammonia). However, despite this, there is good correspondence of all the normalised CD curves.

(3) Line 10, page 2: Authors mentioned the chiroptical measurements in the gas phase had so far been inaccessible for amino acids because of insufficient vapor pressure of the target molecules. Are there any significant improvements in the chiroptical measurement system in the gas phase in your study compared to the past literatures?

Response #3-3:

Thank you very much for this question which motivated us to add the following information to the main text and the supplementary text regarding absorption-based chiroptical measurements in the gas phase:

Main text, introduction, (page 2): With a specifically constructed gas cell coupled to a synchrotron spectropolarimeter to lower the beam divergence that occurs with increased path length, we have now recorded the first CD and anisotropy spectra for enantiomeric pairs of gas phase amino acids [...].

Supplementary File, section S2 (page 2):

Historically, gas phase CD measurements have mostly been limited to volatile molecules, as e.g. carvone (Ballard 1963, Lambert 2012), limonene (Brint 1984), and camphor (Gedanken 1977). Common for these compounds is that they all exhibit a sufficiently high vapor pressure to allow measurements in a relative short pathlength cell of 1 cm operated near room temperature, although longer gas cells have been used to measure the CD of substituted 4-methylcyclohexylidenes (Gedanken 1988). More recently, ion spectroscopy was used to measure gas phase CD spectra of DNA oligonucleotides (Daly 2020) overcoming thermal instabilities by using an electrospray source that on the other hand, however, prevented the measurement of DNA strands in their neutral form. So far, gas phase CD measurements of neutral amino acids have not been reported. The **very low vapor**

pressure of amino acids implies that in order to achieve adequate signals, a combination of elevated temperatures, up to 200 °C, and a long pathlength cell are required.

To overcome these challenges linked to amino acid gas phase measurements, we recorded circular dichroism spectra $CD(\lambda)$, anisotropy spectra $g(\lambda)$, and absorption spectra $A(\lambda)$ of seven pairs of amino acid enantiomers at the ASTRID2 synchrotron radiation source, Aarhus University, Denmark. [...] Using a **synchrotron radiation source, with much lower beam divergence** than found in conventional lamp-based CD spectrometer, allowed for the operation of such a long gas cell.

Additional improvements of our cell – already described in the supplementary text – are:

- In terms of **minimizing amino acid decomposition products**: The gas cell was typically operated under flow conditions, where the valves to the sample and the turbo pump were partially open, to ensure a continuous renewal of the gas in the cell. This was done to avoid a build-up of the products of amino acid decomposition, see sections S2 & S6 for further details.
- In terms of **contamination/cross contamination**: The gas cell has been designed so that it can quickly be taken apart, cleaned, parts replaced if necessary and reassembled. After each cleaning, the entire gas cell was baked off-line under vacuum at 125 °C for a week, to ensure minimal residual water pressure from the clean gas cell.

Additional Supplementary Refs:

Ballard, R. E., Mason, S. F. & Vane, G. W. Circular dichroism of dissymmetric $\alpha\beta$ -unsaturated ketones. *Discuss. Faraday Soc.* **35**, 43–47 (1963).

Lambert, J., Compton, R. N. & Crawford, T. D. The optical activity of carvone: A theoretical and experimental investigation. *J. Chem. Phys.* **136**, 114512 (2012).

Brint, P., Meshulam, E. & Gedanken, A. Excited electronic states of limonene: A circular dichroism and photoelectron spectroscopy study of d-limonene. *Chem. Phys. Lett.* **109**, 383–387 (1984).

Gedanken, A. & Levy, M. New instrument for circular dichroism measurements in the vacuum ultraviolet. *Rev. Sci. Inst.* **48**, 1661 (1977).

Gedanken, A., Duraisamy, M., Huang, J., Rachon, J. & Walborsky, M. Chiroptical properties of chiral olefins. *J. Am. Chem. Soc.* **110**, 4593–4599 (1988).

Daly, S., Rosu, F. & Gabelica V. Mass-resolved electronic circular dichroism ion spectroscopy. *Science* **368**, 1465–1468 (2020).

(4a) Line 23, page 4: Authors calculated the theoretical spectra using several basis sets and compared them with the experimental ones. It is unclear how you decided the best agreement from S9. Did you estimate the spectral differences between experiment and theory, for examples, using the root mean square deviation? **(4b)** Further, are there any reasons that the half-width is 0.4 eV?

Response #3-4a:

Our conclusion on the best agreement between the experimental CD and theoretical spectra focusses mainly on the lowest energetic transition at 240 nm leading to the dominant g band in the spectra of both alanine enantiomers. We addressed this in our first version.

Please see S7: Quantum chemical calculations:

The experimental CD spectra of alanine shows a first negative band at 240 nm. The hybrid functional M06-2X with 54 % of HF exchange was the only method able to predict this negative band at 230 nm. The two other range-separated functionals CAM-B3LYP and ω 97X-D lead to a shift in the negative band at 220 nm (**Supplementary Fig. 9-10**).

However, based on the referee's comment, we calculated the absolute value of local deviation and the of our theoretical spectra:

and added the following information to the Supplementary Information / S7:

Supplementary File, section S9 (page 12):

Experimental and calculated circular dichroism spectra of L-alanine were compared through the determination of the absolute value of local deviation as well as the root-mean-square deviation (RMSD). Overall best correlation was found using the M06-2X method (RMSD = 2.6) followed by the CAM B3LYP (RMSD = 3.2) and W97XD method (RMSD = 3.3). While major differences occur at higher energies for all three DFT methods due to inaccuracies of quantum chemical methods when approaching the ionization threshold as discussed in the main text, the absolute deviation from the experimental spectrum in the wavelength range of 215-250 nm is considerable lower for the calculations performed with M06-2X compared with the two other methods.

Response #3-4b:

While the synchrotron radiation CD spectrometer exhibits a resolution of at least 0.5 nm FWHM, several sources of line broadening result in overall broader widths of CD bands. To consider all possible sources of line broadening as well as finding the best spectral agreement between experimental and theoretical CD spectra without losing any fine structures, a half-width of 0.4 eV was found to provide the best compromise.

(5) Line 18, page 5: Your group used synchrotron radiation for the measurements but the photodegradation products such as water and ammonia have large absorbance in the far UV region, giving a low wavelength limitation in the experimental anisotropy values. These products would be inevitable effect but if we can estimate the partial pressure of water and amino acids using other experimental method or theory, is it possible to remove such lower limit?

Response #3-5:

Again, a very fair comment. We had already described in the Supplementary text S8, the presence of water and ammonia as degradation products mainly due to thermal degradation. To minimize the level

of water coming from the sample, each amino acid had been slowly heated to 80 °C and pumped on for several hours prior anisotropy measurements. The decision to record the anisotropy spectra under flow conditions, i.e. constant intake of amino acid gas into the cell while removing photon- and temperature-induced degradation products simultaneously by carefully pumping on the cell, allowed to keep the level of water and ammonia during the anisotropy measurements low.

However, we have added additional explanations to the Supplementary text S8 as well as Supplementary Fig. 15 to account for the referee's comment on the impact of ammonia/water contamination. Therefore, we have carried out an analysis on the isovaline data – where there is an obvious effect in the *g* spectra for both enantiomers - to better examine the effect of subtraction of the ammonia contribution to the absorbance has. In summary, the effect on the absolute anisotropy values especially at low energies, generally the region with highest *g* values, is rather minor. We therefore believe that a subtraction of water (weak absorption cross section, systematic sample degassing procedure to remove any water, and flow conditions) and ammonia (no spectral features above 215 nm and constant flow conditions to keep ammonia levels low) are not required.

Supplementary File, section S8 (page 20):

“The effect of degradation products on the spectra is most noticeable in the anisotropy spectrum of isovaline (Supplementary Fig. 6b). The spectrum clearly shows fine structure in the 190–210 nm wavelength range which is absent in the CD spectrum (Supplementary Fig. 6a). It originates from ammonia features in the absorbance spectrum used to obtain the anisotropy spectrum.

A scaled ammonia spectrum, shown in **Supplementary Fig. 13**, may be subtracted from the measured absorbance spectrum of isovaline to produce a spectrum free of ammonia features, see **Supplementary Fig. 15a**. Using this as the absorbance spectrum to calculate the anisotropy spectrum of isovaline removes the effect of ammonia on the magnitude of the anisotropy in the 190–210 nm wavelength range as shown in **Supplementary Fig. 15b**. The effect is modest in this wavelength range, with local variations in the anisotropy values of up to 10 %, but no increase in the maximum anisotropy of the band.

Ammonia does not exhibit any spectral features above 215 nm, therefore, the impact on the highest measured anisotropy values for each amino acid is negligible as these generally occur at or above 220 nm. Although water does not have distinct spectral features which would allow subtraction from the measured absorbance spectrum, the conclusion is similar to the one for ammonia: since water doesn't absorb above 185 nm, the highest measured anisotropy values are not affected by water vapour which might be present in the amino acid gas sample. Additionally, the overall cross section for absorption of water³² is smaller than the cross section of ammonia (Limao-Vieira 2019) which, combined with the water degassing procedure described above in this section, further diminish the effect of water on the measured anisotropy spectrum.”

Additional Supplementary Refs:

Limao-Vieira, P., Jones, N. C., Hoffmann, S. V., Dufлот, D., Mendes, M., Lozano, A. I., da Silva, F. F., Garcia, G., Hoshino, M. & Tanaka, H. Revisiting the photoabsorption spectrum of NH₃ in the 5.4 – 10.8 eV energy region. *J. Chem. Phys.* **151**, 184302 (2019).

REVIEWERS' COMMENTS

Reviewer #1 (Remarks to the Author):

The authors have addressed my comments and I now recommend publication.

Reviewer #3 (Remarks to the Author):

Thank you very much for your responses on my comments and questions. These responses are kindly described, and have improved the paper. All of my concerns have been dispelled.